# Cancer Care Team’s Management of Clinical Alerts Generated by Electronically Collected Patient Reported Outcomes: We Could Do Better

**DOI:** 10.3390/ijerph20032001

**Published:** 2023-01-21

**Authors:** Orlando Rincones, Adeola Bamgboje-Ayodele, Anthony Arnold, Geoff P. Delaney, Ivana Durcinoska, Sandra Avery, Tiffany Sandell, Stephen A. Della-Fiorentina, Joanne Pearson, Afaf Girgis

**Affiliations:** 1Ingham Institute for Applied Medical Research, South West Sydney Clinical Campuses, UNSW Medicine & Health, University of New South Wales, Liverpool, NSW 2170, Australia; 2Biomedical Informatics and Digital Health, School of Medical Sciences, Faculty of Medicine and Health, The University of Sydney, Sydney, NSW 2006, Australia; 3Wollongong Hospital, Illawarra Shoalhaven Local Health District, Wollongong, NSW 2500, Australia; 4Liverpool Cancer Therapy Centre, South Western Sydney Local Health District, Liverpool, NSW 2170, Australia; 5School of Medicine, Western Sydney University, Campbelltown, NSW 2560, Australia; 6Macarthur Cancer Therapy Centre, Campbelltown Hospital, Campbelltown, NSW 2116, Australia

**Keywords:** cancer, clinical alerts, eHealth, electronic patient reported outcome measures (ePROMs), patient-centred care, referrals, symptom screening, cancer care coordinators

## Abstract

Electronically administered patient-reported outcome measures (ePROMs) are effective digital health tools for informing clinicians about cancer patients’ symptoms and facilitating timely patient-centred care. This paper describes the delivery of healthcare activities supported by the PROMPT-Care model, including ePROMs generated clinical alerts, cancer care team (CCT) response to alerts, and patients’ perceptions of the CCT response and ePROMs system. This mixed-methods study includes cancer patients from four cancer therapy centres in New South Wales, Australia. Quantitative and qualitative data were collected regarding clinical alert activity, CCT response, and patient perceptions of the CCT responses and ePROMs system. Qualitative data were thematically analysed. Of the 328 participants whose care was informed by the digital health tool, 70.8% (*n* = 233) generated at least one alert during the trial period, with 877 alerts generated in total. Although 43.7% (*n* = 383) were actioned by the CCT, at least 80% of participants found follow-up CCT phone calls beneficial, with multiple benefits confirmed in interviews. The cancer care delivery arm of the PROMPT-Care trial involving clinical alerts to the CCT was positively perceived by most participants, resulting in a diverse range of benefits. However, further work is required, informed by implementation science, to improve the percentage of actioned clinical alerts.

## 1. Introduction

In 2020, over 19 million people were diagnosed with cancer globally [1], while Australia estimates 162,000 new cases in 2021 [2]. With the growing number of cases and improved cancer treatments, the number of cancer survivors is also growing [3]. Cancer prognosis varies depending on tumour site, stage of disease, diagnosis, and treatments available to patients, among others; for instance, 5-year survival rates for breast cancer can reach 80%, while lung cancer is approximately 20% [4]. Cancer patients’ clinical needs also vary depending on a multiplicity of variables; however, research shows that many cancer patients need support on issues such as fatigue, cognitive impairment, fertility, hair loss, mouth health, sexuality and intimacy, taste and smell changes, and peripheral neuropathy, among many others [5]. The growing number of survivors and their various needs place a significant demand on cancer services to meet the needs of survivors. Research has demonstrated that cancer patients experience physical (such as pain and nausea) and psychosocial issues (such as anxiety and depression) before, during, and after treatment [6,7], some of which persist long-term. Patient screening is one of the proactive ways to rapidly identify and appropriately respond to these issues to prevent further escalation and ever-increasing demand on finite healthcare resources.

Electronically completed patient-reported outcome measures (ePROMs) supported by digital health systems [8] are an efficient and effective strategy for alerting clinicians about their cancer patients’ symptoms and unmet needs, in order to facilitate timely care in line with patient-centred care models [9], including health economic endpoints [8] and healthcare quality [10]. There is a growing body of evidence supporting the positive impact of routine collection of ePROMs and timely feedback of their results to the care team, including improvements in patient quality of life [11], management of cancer-related symptoms [12], communication between patient and care team [13], satisfaction with care [12,14], improved cancer survival rates [15], chemotherapy compliance [11], and reduced emergency department (ED) presentations [15,16]. Interestingly, a systematic review of 138 studies found that at least one patient-reported outcome measure (PROM) in 87% of the studies was a significant prognostic factor for overall survival [17], which further supports the importance of PROMs in oncology. However, some argue that, despite increasing use and more solid evidence of PROMs benefits in the past decade, there is still a lack of structure and optimal use of PROMs data [18], with practical issues such as resource availability, reluctance to disrupt clinical workflows, remote data capture, data privacy, and security issues (among others) as relevant challenges for PROMs implementation and sustainability [10,19].PROMPT-Care is an Australian system that facilitates ePROM data collection from cancer patients, data linkage, retrieval, and clinical alerts to support clinical decisions, and patient self-management [20]. When implemented in oncology services, this ePROMs system was demonstrated to be feasible to implement and acceptable to patients and healthcare professionals [21], and when used to inform the routine care of patients with a wide range of solid tumours undergoing active treatment or follow-up care, it led to significantly reduced ED presentations, compared with a matched control group [16]. The two key components of the efficacious PROMPT-Care intervention are (a) the patient support self-management arm and (b) the cancer service driven arm, involving clinical alerts to health care providers. Data relating to the patient self-management arm is published elsewhere [22]. Since a critical feature of efficacious ePROM interventions is a timely feedback and response to patient reports, this paper focuses specifically on the cancer service arm of the PROMPT-Care trial, describing the activities relating to clinical alerts and cancer care team (CCT) response to these alerts and patients’ perceptions of the CCT’s response.

## 2. Materials and Methods

### 2.1. Study Design and Participants

A mixed-methods study was conducted during the PROMPT-Care 2.0 pragmatic trial [16], which took place from April 2016 to October 2018 and included cancer patients from four cancer therapy centres from two different local health districts (LHD). The South Western Sydney LHD (the two largest and busiest cancer care centres were included: Liverpool Cancer Therapy Centre and Macarthur Cancer Therapy Centre) is a socially, economically, culturally, and linguistically diverse community in the Sydney Greater Region. A significant proportion of its population was born overseas (43%), compared to the average in the state of New South Wales (25.7%), and nearly half (45%) of South Western Sydney residents speak a language other than English at home [23]. The Illawarra and Shoalhaven LHD (the two largest and busiest centres were included: Illawarra Cancer Care Centre and Shoalhaven Cancer Care Centre) is a district whose communities have a substantial proportion of people aged 75 years and older (8.3%), when compared to the NSW average (6.74%) [24]. In contrast with South Western Sydney LHD, only 18% of the Illawarra and Shoalhaven LHD population was born overseas [24].

The eligibility criteria were (a) aged at least 18 years old, (b) receiving active treatment (i.e., chemotherapy or radiotherapy) or follow-up care for a solid tumour, (c) ability to complete ePROMs in English, (d) ability to provide informed consent, and e) access to the internet and an email address. The South Western Sydney Local Health District Human Research Ethics Committee approved this study (reference HREC/15/LPOOL/287) and patients provided signed informed consent. Ethics, data privacy, and confidentiality followed the principles of the Declaration of Helsinski [25] and the standards of the Australian Code for the Responsible Conduct of Research [26].

### 2.2. The PROMPT-Care 2.0 Intervention–Clinical Alerts

As previously reported [16,21], the PROMPT-Care system involved cancer patients completing ePROMs monthly, including the distress thermometer (DT) and associated problem checklist [27], the Edmonton symptom assessment scale (ESAS) [28], and the supportive care needs survey-screening tool 9 (SCNS-ST9) [29], with responses automatically stored in participants’ electronic medical records (eMR). In real-time, algorithms identified ePROM items that breached pre-determined alert thresholds [30], generating (a) an email to patients with links to relevant online self-management resources [22], and (b) a ‘clinical alert’ email containing the medical record number (MRN) of the concerned patient (Figure 1), prompting the care team to review the report highlighting any items breached on two consecutive ePROMs and the recommended care to address these, as well as a longitudinal report of all ePROM items over time [20] (see Figure 2 and Figure 3). Examples of these reports have been published previously [20]. Health care providers who were part of the cancer care team (CCT) were trained on how to access the ePROM reports, consider the recommended actions [30], alongside their clinical judgement, and record final actions taken in patients’ eMRs. ePROMs items were grouped into five domains: physical, emotional, practical, social/family, and maintenance of wellbeing, with a total of 15 clinical alert recommendations mapped to these 61 items [30].

### 2.3. Procedures

Treating clinicians from participating sites pre-screened their patients for potential eligibility, and eligible patients were then invited to participate in the study by a member of the research team or nursing staff. Participants received the study information via mail or in-clinic and were contacted via phone to answer any of their questions and check interest in participating. Consenting patients received an email with the link to complete the first ePROMs survey, and subsequent surveys were sent monthly, with a reminder email sent a week later if the survey had not yet been completed. Staff reviewing the clinical alerts were instructed to call the patient within 2–3 days of an alert being generated and record their actions in the ‘Chart notes’ section of the oncology information system (eMR system for oncology patients (MOSAIQ^®^)), as per routine administrative procedures. Staff were encouraged to mention ‘PROMPT-Care’ in their ‘Chart notes’, in order to determine that the care provided was elicited by the review of the ePROM reports generated as part of the research trial.

To evaluate participants’ views about the PROMPT-Care system, patients received evaluation surveys three, six, and nine months after starting the intervention and were invited to participate in a phone interview regarding their more in-depth views of the system.

### 2.4. Outcome Measures

#### 2.4.1. Clinical Alert Activity and Cancer Care Team Response

Data were extracted from the PROMPT-Care system on (a) the number of clinical alerts generated across all participants, and (b) the most commonly breached items from the DT and checklist [27], ESAS [28], and SCNS-ST9 [29]; notes within MOSAIQ^®^ were interrogated to document CCT responses to clinical alerts.

#### 2.4.2. Patient Perceptions of the CCT Responses

Study-specific evaluation surveys (44 items, some of which have been reported elsewhere [22]) were sent to participants at 3 months, 6 months, and 9 months to determine their perceptions of the CCT follow-up, relating to clinical alerts. As detailed in the results section, the nine questions related to the clinical alerts specifically used either dichotomous or likert response options (strongly agree, agree, neutral, strongly disagree, disagree). Qualitative semi-structured interviews were also conducted to gain a deeper understanding of the patients’ experiences (see Appendix A).

### 2.5. Analysis

Quantitative data were analysed descriptively and reported as frequencies, means, and standard deviations. Qualitative analysis of audio-recorded interviews was based on Braun and Clarke’s thematic analysis [31]. Verbatim transcriptions of semi-structured interviews were analysed to better understand participants’ views around the acceptability and perceived impact of CCT follow-up to clinical alerts. The average length of the interviews was 41 min (range 23 to 60 min). The core research team developed the interview schedule (See Appendix A). Two researchers conducted the interviews (OR and MG), and two researchers (OR and SB) coded the transcripts and identified emerging themes. Discrepancies were resolved through discussion and consensus.

## 3. Results

### 3.1. Participant Characteristics

A total of 328 participants received the intervention. Participants were 62.4 years old on average and included 59.5% female. The most common tumour sites were breast (*n* = 132, 40.2%) and prostate cancer (*n* = 51, 15.6%), and 42.46% were receiving active treatment (chemotherapy and/or radiotherapy). Refer to Table 1 for other participant characteristics.

### 3.2. Clinical Alert Activity and Response

#### 3.2.1. Clinical Alerts and Breached Items

Of 328 participants, 70.8% (*n* = 233) had at least one alert generated during the trial period, with a total of 877 clinical alerts generated from April 2016 to October 2018. On average, 31 clinical alerts were generated every month (SD = 16.8; range = 2–78 alerts per month). A total of 3693 items were breached across all alerts, with an average of 4.2 items per alert (SD = 4.55; range = 1–33 items). Out of 877 clinical alerts, the most commonly breached items were fatigue (29.6%, *n* = 260), tiredness (29.2%, *n* = 256), tingling in hands and feet (26.4%, *n* = 232), worry (24.5%, *n* = 215), and wellbeing (19.1%, *n* = 168).

#### 3.2.2. Cancer Care Team Responses to Clinical Alerts

Figure 4 shows a workflow of responses to clinical alerts. Of the 877 alerts generated, 383 were explicitly recorded as actioned (43.7%). A total of 496 actions were recorded in eMRs, which could be directly linked to the alerts, with more than one action possible per alert (e.g., two phone calls). The most common actions included discussing the alert with the patient (*n* = 302, 61%), contact attempted but unable to reach the patient (*n* = 111, 22%), and alert reviewed but, following clinical assessment of patient notes, patient contact was deemed unnecessary (*n* = 83, 17%) (e.g., ongoing issue previously discussed with patient and under management).

Of the 302 alerts discussed with patients, 43% (129) did not require any further action, 32% (98) resulted in providing advice, and 25% (75) resulted in offering one or more referrals, with a total of 86 referrals offered (see Table 2). The referral acceptance rates are displayed in Table 2. The number of clinical alerts being actioned differed among study sites: Site 1 actioned 79.3% (176) of alerts, Site 2 actioned 48.7% (56), Site 3 actioned 33.5% (55), and Site 4 actioned 25.5% (96) of alerts.

#### 3.2.3. Participant Evaluation Surveys

The 3-month evaluation survey was completed by 221 participants, 78 participants completed the 6-month and 93 completed the 9-month surveys. Most participants did not recall receiving a phone call from the CCT in response to their PROMPT-Care surveys (67.4% at 3-months, 73.1% at 6-months, and 72% at 9-months). Of those who remembered receiving a phone call, at least 80% found these phone calls beneficial across all time points, and 66.7% or more felt that their nursing team provided them with enough information and support during the phone call. Additionally, 57% or more of participants expressed that the phone call helped them to deal with current issues. When asked about linkage with appropriate support services based on their reported needs, at least 62% of respondents at 3-months and 9-months agreed that the phone call facilitated this; however, only 38.1% agreed with this at 6-months. Only 8.3% or fewer participants would have preferred not to receive the phone call, and 4.8% or less found the phone call bothersome across all timepoints. Most respondents (57% or more) considered that the phone call resulted in better communication with their Cancer Care Centre on how they were feeling (see Table 3).

### 3.3. Qualitative Findings

The thematic analysis revealed an overarching theme and four main themes, as reflected in Figure 5. See Appendix B to access the further supportive quotes to main themes and subthemes presented in the thematic map.

#### 3.3.1. Overarching Theme: Multiple Perceived Benefits from the Cancer Care Team Call for Most Patients

The majority of participants reported some degree of benefit from the call received from the CCT as a result of the clinical alerts generated from their ePROMs survey. Participants expressed satisfaction with the PROMPT-Care system and the phone call from their CCT, citing perceived benefits, such as feeling being cared for and checked on by the CCT, and valuing the referrals made. Such benefits made them feel valued by their CCT and the health system, and some had a sense that PROMPT-Care had a real impact on their care and not just feeling ‘like a number’. This quote illustrates a participant’s views about receiving a quick phone call after completing the PROMPT-Care assessment:


*I suppose there are so many people like me in the hospital system. It’s like you do feel you become a number. It just made me feel important, like I mattered, my treatment mattered, and care mattered… not just paperwork and then filed. There was an actual result from someone rang me up and quite quickly actually. I was very impressed with that. I think it is important that people think that their treatment—it’s not just a number. They’re treating people not just numbers.*
(L136)

#### 3.3.2. Main Themes

Feeling cared for

The most common benefit reported was a feeling of being cared for. Participants felt valued and cared about by the CCT. The phone call reassured participants and made them feel that there was continuity in care from their CCT even after treatment.

Valuable referrals to HCPs

Patients valued the referrals to allied health services, such as dieticians, clinical psychologists, and other support services, and felt that they pointed them in the right direction to deal with their issues.

Other benefits

Interviews revealed other benefits resulting from the CCT clinical alert follow-up phone calls. For example, a few participants mentioned that they found the phone calls to be helpful follow-ups in between oncology appointments and improved their communication with the CCT to discuss their issues. One participant highlighted the benefit of the phone call to address her problems, without ‘the stress of attending a consultation’ at the cancer therapy centre.

Necessity of the PROMPT-Care phone call

Although most participants found the CCT phone call beneficial, a few felt it was irrelevant to them, but still appreciated the call and understood why they received it. Some participants thought that phone call was not needed, as they felt they did not have too many issues or their issues were not chronic/acute. Others reported that they might have answered the survey questions ‘incorrectly’, triggering an unnecessary call. They expressed that the phone call could benefit other patients with more issues, slower recovery, or patients with little social support.

Some participants felt surprised about receiving the phone call following the completion of a PROMPT-Care survey. Additionally, there were mixed views about including a question at the end of the survey that asks whether the patient wants to be contacted or not regarding issues mentioned in the survey. Some participants explained that it is ‘up to the patient’ if they need a call or not. However, others believed that such a question might not be useful because some patients might opt out from the call when they actually need the support from the call.

## 4. Discussion

This paper aimed to describe the activities relating to clinical alerts, the cancer care team responses to these alerts, and patients’ perceptions of the ePROMs system. A large proportion of participants (70.8%) breached ePROMs thresholds on two or more consecutive occasions, breaching an average of four items (out of 61 potential items) on each assessment. The most commonly breached items were fatigue, tiredness, tingling in hands and feet, worry, and wellbeing, which are consistent with the current literature [32], reiterating the ongoing cancer-related burden in this population [6,7].

The overall proportion of reviewed reports and alerts being actioned (43.7%) was lower than expected, although it is important to note that the rates of action varied widely across study sites, ranging from 25.5% to 79.3%. This differs from the eRAPID study [33], an ePROMs system for cancer patients going through chemotherapy, which showed that the proportion of clinicians reviewing patients’ ePROMs was 81.4% (however, there is no systematic reporting of what actions were taken after ePROMs review). A possible explanation for the lower than expected responses to alerts might be due to difficulties in changing current clinical workflows, as reported in the literature [10]. When the care team actioned alerts, this generally resulted in them telephoning the patient and discussing the reason for the alert (61%) and offering a referral if required (25%) or providing advice (32%). Patient evaluation surveys revealed that the vast majority who remembered receiving a phone call from the CCT found it beneficial (>80%) and believed that the phone call improved their communication with their cancer care team (>57%), which is consistent with the multiple perceived benefits interviewees noted and the low proportions of patients who found the phone call bothersome (≤4.8%).

Many of the alerts discussed were judged by the CCT as not requiring any further action (43%), due to ongoing issues having already being addressed by the care team or by the patient independently. This highlights the importance of ePROMs reports being reviewed in conjunction with patients notes and clinical judgement. Our CCTs’ response to alerts is consistent with the RELIEF study [34], an ePROMs intervention of daily assessments in palliative care cancer patients, which found that nearly half of the alerts generated required clinical intervention. Gaining a better understanding of the issues that generated alerts, as well as a review of the actions in response to clinical alerts, provides important guidance to inform potential revisions to the ePROM thresholds in the algorithms that generate the clinical alerts. This may go some way to reducing the overall number of “unnecessary” alerts, addressing the challenge of varied access to resources, such as oncology nursing at different sites, but also the overall support and uptake of the PROMPT-Care system in practice.

Our system did not include a requirement to formally acknowledge an alert and record an action (including “No action taken”) in the eMR. Hence, the reported action rate may be an underestimate. The reporting of actions could potentially be greatly increased through IT solutions, such as adding a drop-down menu that needs to be completed by the CCT (as per other items in the eMR), potentially providing a much more accurate picture of CCT response to patients’ reported concerns. With monitoring and feedback demonstrated as an effective strategy for enhancing compliance with recommended practice [35,36], the addition of daily or weekly monitoring of actions and review in staff meetings might also encourage the more systematic actioning of alerts.

The level of implementation readiness at each cancer service, including the ongoing challenge of stretched or limited staffing, may also have contributed to the less-than-optimal actioning of clinical alerts. An estimated 31 clinical alerts were generated monthly across all four hospitals. Whilst this number of alerts distributed across four different cancer sites may not be particularly burdensome for the cancer services overall, it may have impacted certain tumour-specific teams differently. Each alert was directed to specific care teams, and some care teams, particularly the breast and prostate cancer nurses, received a higher number of alerts because of the larger study samples from these tumour groups.

One quarter of the CCT phone calls to patients resulted in referrals to other services, with clinical psychology being the most common allied health referral offered (*n* = 55), followed by dietetics (*n* = 11) and social work (*n* = 9). Most of the clinical psychology service referrals (67.3%) and social work referrals (77.8%) were declined by patients. This is consistent with published evidence showing a low uptake of psychological services [37,38], sometimes because patients might not perceive a need for such support, or a stigma associated with use of these services [39,40]. In contrast, referrals to the dietetics service were accepted by 72.7% of participants, which might reflect the level of openness to referral to these services, the digestive side effects of some cancer treatments [41] and the awareness of the role of diet in preventing cancer [42].

Most participants did not recall receiving a phone call from the CCT in response to the PROMPT-Care surveys. The potential reasons for this include (a) participants did not report issues on their surveys and, hence, did not receive calls, (b) the CCT did not explicitly mention their PROMPT-Care survey results during phone calls, (c) participants were not reached by the CCT (either because alerts were not actioned, alerts were reviewed but clinical assessment did not suggest the need for a call, or staff was unable to reach the patient), or (d) recall bias.

Very few participants did not find benefits from the phone calls (≤5%) or were neutral about the call (≤17%). This aligns with a proportion of interviewed patients expressing that the PROMPT-Care phone call might be unnecessary because they were not experiencing many (or acute) issues. Interestingly, some participants appreciated the phone call, but noted that it might be more useful for those with more issues or slower recovery. ePROMs systems can readily be adapted to inform stepped care or a more tailored model of care, based on individual patient responses to their ePROMs, including how much direct unscheduled contact is made by the care team. However, care is required to ensure that patients are not declining care or contact because they do not want to ‘bother’ their providers or they feel that their issues are not significant, compared to other patients, particularly as timely identification and response to issues prevents increased deterioration and emergency department presentations [15,16].

This study has a number of limitations. A large number of clinical alerts were not actioned by the CCT, especially in one recruitment site. While we reviewed patients’ eMR notes to identify actions taken after clinical alerts were generated, some actions may have been missed, due to human error, actions not being recorded by the CCT, or no explicit mention of ‘PROMPT-Care’ in the note. CCT feedback regarding their use of the system, barriers, and facilitators could also have added important insights to the study.

## 5. Conclusions

Our examination of the use of PROMPT-Care, a digital health system implemented in four cancer centres, regarding clinical alert activity, CCT responses, and patient perceptions of the CCT responses, shows promising results regarding the feasibility of such digital systems for delivery of routine healthcare. The care delivery arm of the PROMPT-Care trial involving the automatic generation of clinical alerts to the CCT was positively perceived by most participants, resulting in a diverse range of benefits. However, further work is required, informed by implementation science frameworks, to improve the uptake of referrals from the patients’ perspective and the percentage of actioned clinical alerts from the clinicians’ perspective. From a translational point of view, immediate or timely response (up to 2–3 days) to clinical alerts generated by ePROMs is ideal, and a standard and friendly system to record actions is needed. For optimal uptake within the clinical workflows, health managers, along with their clinical and IT teams, should determine the best and easiest ways to achieve this.

This study also provides an important platform for review of action thresholds, which can be adapted based on local populations and available resources. Our study contributes to the development and implementation of digital strategies and technology to effectively use ePROMs in routine care, an issue raised in recent literature [43,44,45]. This work also reinforces the importance of a staged introduction of ePROMs into routine care, taking into consideration important local contexts and resources (human and financial).

## Figures and Tables

**Figure 1 ijerph-20-02001-f001:**
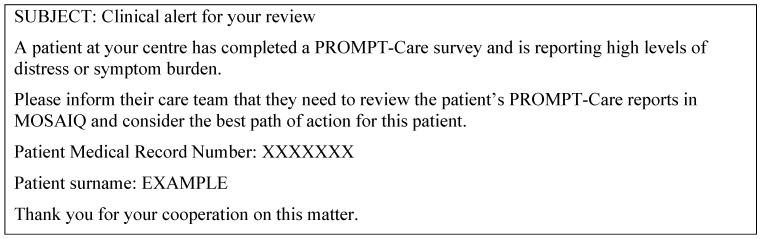
Clinical alert E-mail.

**Figure 2 ijerph-20-02001-f002:**
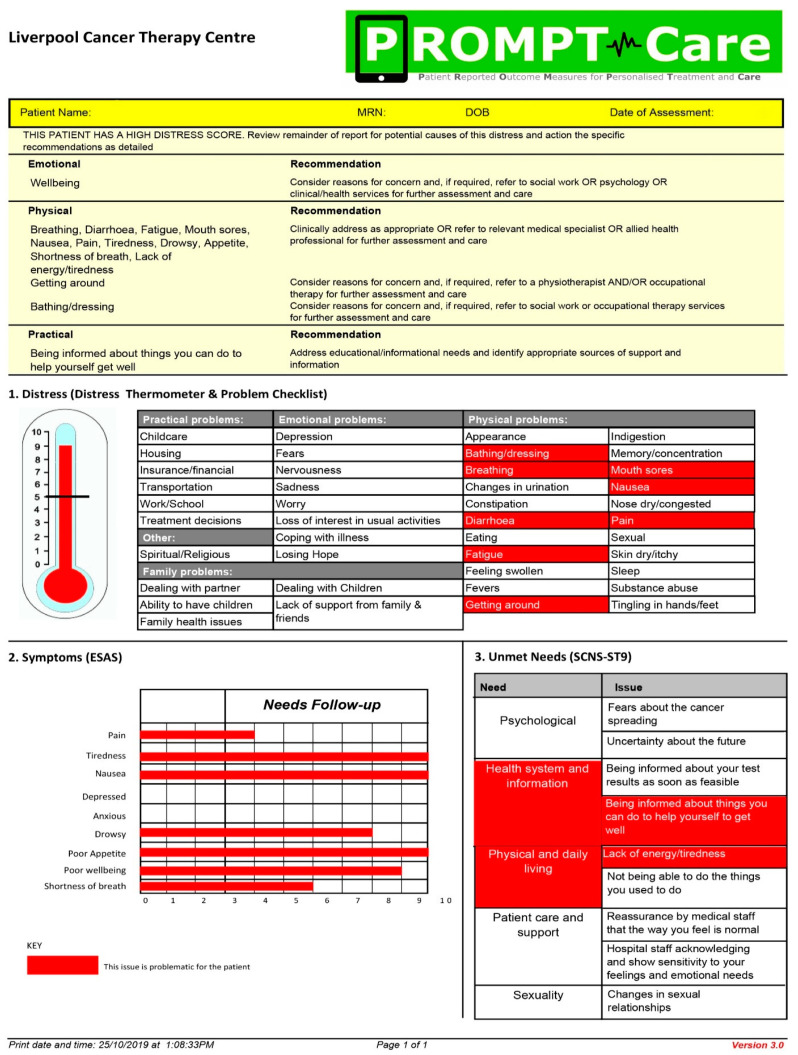
Sample clinical feedback report. Originally published in: [16].

**Figure 3 ijerph-20-02001-f003:**
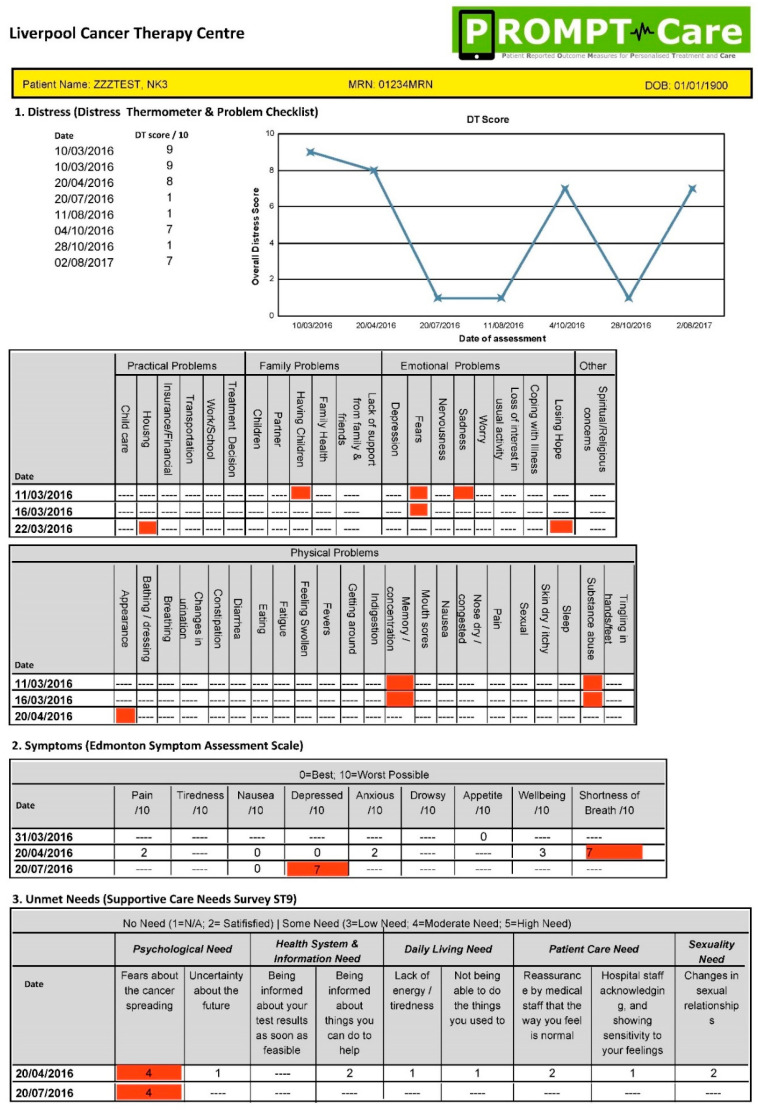
Sample longitudinal feedback report. Originally published in: [20].

**Figure 4 ijerph-20-02001-f004:**
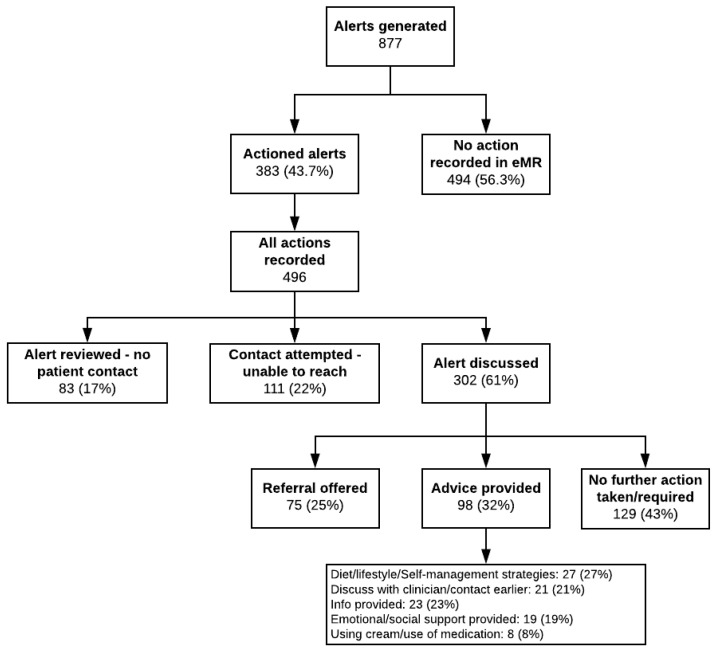
Clinical alerts generated and cancer care team responses. Note 1: A clinical alert might generate multiple actions. For instance, two separate phone calls on different days (in the first phone call, the CCT provided information, while during the second phone call, a referral was offered).

**Figure 5 ijerph-20-02001-f005:**
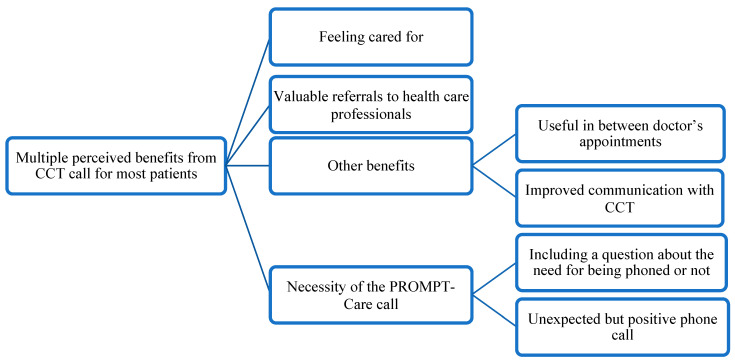
Thematic map.

**Table 1 ijerph-20-02001-t001:** Participant Demographic Characteristics.

Characteristics	*n* (%)
Age (years), mean (range)	62.4 (25–86)
Sex, *n* (%)	
Male	133 (40.6)
Female	195 (59.5)
Site of cancer	
Breast	132 (40.2)
Prostate	51 (15.6)
Colorectal	37 (11.3)
Respiratory	29 (8.8)
Gynaecological	16 (4.9)
Upper gastrointestinal	15 (4.6)
Skin	11 (3.4)
Oral	10 (3.1)
Other	27 (8.2)
Stage of disease	
0/I	66 (22.1)
II	90 (27.4)
III	57 (17.4)
IV	80 (24.4)
Missing	35 (10.7)
Treatment status	
Active treatment ^b^	139 (42.4)
Follow-up care	189 (57.6)
Socioeconomic status (IRSD) ^c^	
1	54 (16.4)
2	97 (29.6)
3	52 (15.9)
4	35 (10.7)
5	90 (27.4)
Relationship status ^a^	
Single	71 (23.1)
Partnered	236 (76.9)
Education status ^a^	
High school or less	122 (39.7)
Post-secondary education	185 (60.3)
Employment ^a^	
Employed	129 (42)
Retired	155 (50.5)
Other	23 (7.5)

^a^ Some level of missing data. ^b^ Chemotherapy, radiotherapy, or both. ^c^ IRSD: Index of relative socioeconomic disadvantage. 1 = most disadvantaged; 5 = least disadvantaged.

**Table 2 ijerph-20-02001-t002:** Acceptance rates of referrals made following review of clinical alerts.

Referral	Total Offered	Accepted
Clinical psychology	55	18 (32.7)
Dietetics	11	8 (72.7)
Social work	9	2 (22.2)
Physiotherapy	3	3 (100)
Prostate cancer nurse	3	3 (100)
Medical oncology	2	2 (100)
Unknown *	3	0
Total	86	36 (41.9)

* Electronic Medical Record shows that the patient was ‘referred, but declined’ without mention of which service.

**Table 3 ijerph-20-02001-t003:** Evaluation survey responses.

	3 Months		6 Months		9 Months	
	(*n* = 221)		(*n* = 78)		(*n* = 93)	
	*n*	%	*n*	%	*n*	%
Do you recall receiving a phone call from the nursing team at your Cancer Care Centre in response to your PROMPT-Care assessments?						
* Yes*	72	32.6	21	26.9	26	28
I found receiving a phone call from the nursing team at my Cancer Care Centre beneficial						
* Strongly Agree/Agree*	58	80.6	17	81.0	22	84.6
* Neutral*	12	16.7	3	14.3	3	11.5
* Strongly Disagree/Disagree*	2	2.8	1	4.8	1	3.8
The phone call from the nursing team helped me deal with ongoing problems I was experiencing						
* Strongly Agree/Agree*	46	63.9	12	57.1	17	65.4
* Neutral*	22	30.6	7	33.3	8	30.8
* Strongly Disagree/Disagree*	4	5.6	2	9.5	1	3.8
Contact with my Cancer Care Centre linked me back to appropriate support services to help with my problems (e.g., Dietitian, Psychologist)						
* Strongly Agree/Agree*	45	62.5	8	38.1	18	69.2
* Neutral*	22	30.6	9	42.9	7	26.9
* Strongly Disagree/Disagree*	5	6.9	3	14.3	1	3.8
The nursing team provided me with enough information and support						
* Strongly Agree/Agree*	59	81.9	14	66.7	21.0	80.8
* Neutral*	12	16.7	6	28.6	4.0	15.4
* Strongly Disagree/Disagree*	1	1.4	1	4.8	1.0	3.8
I would have preferred not to receive a phone call from the nursing team in response to my PROMPT-Care assessments						
* Strongly Agree/Agree*	6	8.3	1	4.8	2	7.7
* Neutral*	8	11.1	3	14.3	4	15.4
* Strongly Disagree/Disagree*	58	80.6	17	81	20	76.9
I would have preferred to be able to contact the nursing team when I needed assistance						
* Strongly Agree/Agree*	26	36.1	7	33.3	5	19.2
* Neutral*	22	30.6	7	33.3	11	42.3
* Strongly Disagree/Disagree*	23	31.9	7	33.3	10	38.5
I found receiving a phone call from the nursing team bothersome						
* Strongly Agree/Agree*	3	4.2	1	4.8	1	3.8
* Neutral*	8	11.1	1	4.8	2	7.7
* Strongly Disagree/Disagree*	61	84.7	19	90.5	23	88.5
The phone call from the nursing team facilitated better communication with my Cancer Care Centre on how I was doing						
* Strongly Agree/Agree*	45	62.5	14	66.7	15	57.7
* Neutral*	20	27.8	5	23.8	11	42.3
* Strongly Disagree/Disagree*	5	6.9	2	9.5	-	-

Note: some level of missing data as participants skipped some questions.

## Data Availability

The data that support the findings of this study are available on request from the corresponding author. The data are not publicly available, due to privacy or ethical restrictions.

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
