# Peer review of "Cancer Care Team’s Management of Clinical Alerts Generated by Electronically Collected Patient Reported Outcomes: We Could Do Better"

_ijerph, 2023, doi:10.3390/ijerph20032001_

Round 1

Reviewer 1 Report

This paper is about healthcare activities in Australia, Sydney Region, and New South Wales. The authors show the model of electronic surveys for cancer patients and the interaction of these patients with cancer care teams. The manuscript is well-written, and clearly explains methods, results, and conclusion.
In table 1., the sum of patients in column Socioeconomic status is 329 (it should be 328).

Author Response

In table 1., the sum of patients in column Socioeconomic status is 329 (it should be 328).

Thank you very much for your positive feedback. We have fixed the typo in Table 1.

Reviewer 2 Report

In the present manuscript entitled “Cancer Care Team’s Management of Clinical Alerts Generated by Electronically Collected Patient Reported Outcomes: We Could Do Better” the authors have assessed the value of electronically submitted patient monitoring, regarding cancer management.

This is a well-designed study, however, there are a number of minor issues that authors have to address to be able to publish their results.

1.       Please provide more information on cancer regarding prognosis and clinical current need.

2.       Please improve the quality of Figures 2 & 3.

3.       The authors should discuss more the translational impact of their findings for disease management.

Author Response

  1. Please provide more information on cancer regarding prognosis and clinical current need.

- Thanks for the suggestion. We will add the following sentences to the introduction: “Cancer prognosis varies depending on tumour site, stage of disease, diagnosis and treatments available to patients, among others; for instance, 5-year survival rates for breast cancer can reach 80%, while lung cancer is approximately 20%. Cancer patients’ clinical needs also vary depending on a multiplicity of variables; however, research shows that many cancer patients need support on issues such as fatigue, cognitive impairment, fertility, hair loss, mouth health, sexuality and intimacy, taste and smell changes, peripheral neuropathy, among many others”.

  1. Please improve the quality of Figures 2 & 3.

- Thanks for pointing this out; we agree. The figures in the word document display less quality. However, the figure files attached to the submission have better quality. We have uploaded better quality TIFF figures. Note to the editor: please use the figure files attached.

  1. The authors should discuss more the translational impact of their findings for disease management.

- Thanks for raising this point. We will add the following sentences to the manuscript’s conclusion: “From a translational point of view, immediate or timely response (up to 2-3 days) to clinical alerts generated by ePROMs is ideal and a standard and friendly system to record actions is needed. For optimal uptake within the clinical workflows, health managers, along with their clinical and IT teams, should determine the best and easiest ways to achieve this.”

Reviewer 3 Report

This paper explores the efficacy of electronically administered patient-reported outcome measures (ePROMs) as digital health tools for informing clinicians about cancer patients' symptoms and facilitating timely, patient-centred care. The study found that out of 328 participants, 70.8% generated at least one clinical alert, which were responded to by the Cancer Care Team in 43.7% of cases. While the study has a number of limitations, including a large number of clinical alerts that were not actioned by the CCT. Surveys and interviews showed that most participants found the follow-up phone calls beneficial, leading to a diverse range of positive outcomes.

Minor comments:

1. figure 2, 3 has poor resolution. Please use high resolution ones.

2. please centralize your Tables

Author Response

  1. Figure 2, 3 has poor resolution. Please use high resolution ones.

- Thanks for pointing this out; we agree. The figures in the word document display less quality. However, the figure files attached to the submission have better quality. We have uploaded better quality TIFF figures. Note to the editor: please use the figure files attached.

  1. Please centralize your Tables.

- Tables have now been centralized.

Reviewer 4 Report

Some points need to be further clarified:

This is nice and meaningful job, but a few points need further clarification:

1.        The introduction can be written more comprehensively and substantially.

2.        Unfortunately, some significant work is not considered nor cited, such as:

https://doi.org/10.1159/000519151

http://dx.doi.org/10.1136/bmjqs-2018-008426

3.        It is necessary to establish some indicators to evaluate We Could Do Better.

4.        Some remarks on the main results would be necessary and helpful.

Author Response

  1. The introduction can be written more comprehensively and substantially.

- Thanks for this comment. We struggled to respect to the word limit. We have now added the following sections to the introduction:

“Cancer prognosis varies depending on tumour site, stage of disease, diagnosis and treatments available to patients, among others; for instance, 5-year survival rates for breast cancer can reach 80%, while lung cancer is approximately 20%. Cancer patients’ clinical needs also vary depending on a multiplicity of variables; however, research shows that many cancer patients need support on issues such as fatigue, cognitive impairment, fertility, hair loss, mouth health, sexuality and intimacy, taste and smell changes, peripheral neuropathy, among many others”.

‘Interestingly, a systematic review of 138 studies found that at least one patient-reported outcome measure (PROM) in 87% of the studies was a significant prognostic factor for overall survival, which further supports the importance of PROMs in oncology. However, some argue that despite increasing use and more solid evidence of PROMs benefits in the past decade, there is still a lack of structure and optimal use of PROMs data, with practical issues such as resource availability, reluctance to disrupt clinical work-flows, remote data capture, data privacy and security issues (among others) as relevant challenges for PROMs implementation and sustainability.’

  1. Unfortunately, some significant work is not considered nor cited, such as:

 https://doi.org/10.1159/000519151

 http://dx.doi.org/10.1136/bmjqs-2018-008426

- Apologies for not including these references. We have now included them in the Introduction section (second paragraph) and Discussion (second paragraph).

  1. It is necessary to establish some indicators to evaluate “We Could Do Better”.

- Thanks for bringing this up. The ‘We could do better’ statement in the manuscript’s article attempts to reflect on the overall belief that the management of clinical alerts from ePROs can be improved. As we have transparently reported that a number of clinical alerts were not actioned (or at least there is no documented evidence of being actioned), we believe this could be improved. We believe that the discussion points to a number of areas where we can indeed to better, hence, unless strongly recommended by the Editor, we believe the current title is appropriate.

  1. Some remarks on the main results would be necessary and helpful.

- The main results and implications of these have been discussed in the Discussion section. It’s not entirely clear what further remarks this reviewer is seeking, as none of the other reviewers have commented on this.